# DISTFLASHATTN: Distributed Memory-efficient Attention for Long-context LLMs Training

**Dacheng Li** [*b]
dacheng177@berkeley.edu

**Rulin Shao** [*w]
rulins@cs.washington.edu

**Anze Xie** [s]
a1xie@ucsd.edu

**Eric P. Xing** [c]
ericxing@cs.cmu.edu

**Xuezhe Ma** [u]
xuezhema@isi.edu

**Ion Stoica** [b]
istoica@berkeley.edu

**Joseph E. Gonzalez** [b]
jegonzal@berkeley.edu

**Hao Zhang** [s]
haozhang@ucsd.edu

[b] UC Berkeley    [w] University of Washington    [s] UCSD    [c] CMU    [u] USC

## Abstract

FlashAttention (Dao, 2023) effectively reduces the quadratic peak memory usage to linear in training transformer-based large language models (LLMs) on a single GPU. In this paper, we introduce DISTFLASHATTN, a distributed memory-efficient attention mechanism optimized for long-context LLMs training. We propose three key techniques: token-level workload balancing, overlapping key-value communication, and a rematerialization-aware gradient checkpointing algorithm. We evaluate DISTFLASHATTN on Llama-7B and variants with sequence lengths from 32K to 512K. DISTFLASHATTN achieves 8× longer sequences, 4.45 – 5.64× speedup compared to Ring Self-Attention, 2 – 8× longer sequences, 1.24 – 2.01× speedup compared to Megatron-LM with FlashAttention. It achieves 1.67× and 1.26 – 1.88× speedup compared to recent Ring Attention and DeepSpeed-Ulysses. Codes are available at https://github.com/RulinShao/LightSeq.

## 1 Introduction

Large language models (LLMs) capable of processing long context have enabled many novel applications, such as generating a complete codebase (Osika, 2023) and chatting with long documents (Li et al., 2023). Yet, training these LLMs with long sequences significantly increases activation memory footprints, posing new challenges.

Contemporary approaches to manage the high memory demands of long-context LLMs training involve either reducing activation memory on a single device or partitioning and distributing the sequences across multiple devices. Memory-efficient attention (Dao et al., 2022; Dao, 2023; Rabe & Staats, 2021) represents the former, which reduces the peak memory usage of attention operations on a single device. Despite their effectiveness, the absence of a distributed extension limits their application to sequence lengths that a single device can accommodate. Naively combining it with existing tensor or pipeline parallelisms (Shoeybi et al., 2019)) leads to excessive communication (§ C) and cannot scale with sequence length (§ 4). On the other hand, sequence parallelism systems, Ring Self-Attention (Li et al., 2021) and Ring Attention (Liu et al., 2023), distribute the activations of a long sequence across multiple devices, but they lack support for memory-efficient attentions (e.g., FlashAttention) or scheduling optimizations, making them inefficient in training long sequences (§ 4.4).

---

*Authors contributed equally.

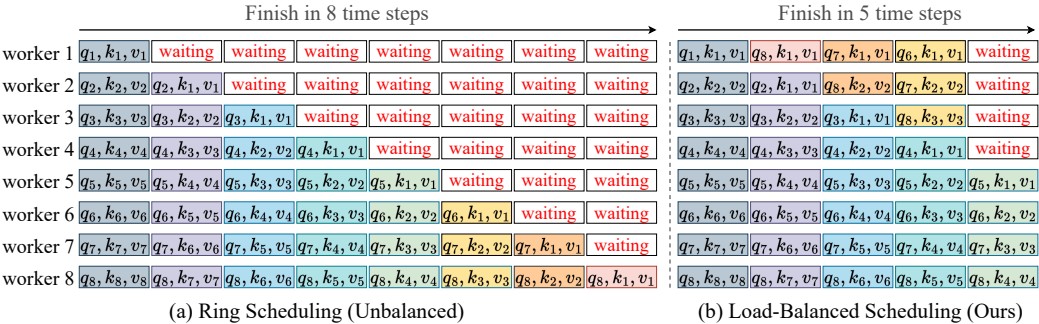

Figure 1: Per-worker workload at different time steps in (a) ring scheduling (Li et al., 2021) and (b) the proposed load-balanced scheduling in an 8-worker scenario. The causal attention introduces a quadratic work dependency on the prefix of each token, where workers assigned earlier tokens remain idle while waiting for workers with later tokens. The idle fraction of the ring scheduling is $\frac{P^2-P}{2P^2}$, asymptotically $\frac{1}{2}$ when scaling to more number of workers. The idle fraction of the proposed load-balanced scheduling is $\frac{1}{2P}$ when $P$ is even and 0 when $P$ is odd, asymptotically 0 when scaling to a larger number of workers.

This paper introduces DISTFLASHATTN to extend the advantages of FlashAttention (Dao, 2023) to the distributed setting while maintaining high GPU utilization and low communication overhead. DISTFLASHATTN efficiently distributes token chunks across multiple devices, while maintaining the IO-aware benefits of memory-efficient attention. We identify three key challenges in achieving high GPU utilization on distributed FlashAttention design for long-context LLMs and propose three optimizations to addgress them.

The first challenge is the token-level workload imbalance caused by causal language modeling. As shown in Figure 1 (a), the causal attention introduces a quadratic work dependency on the prefix of each token. This leads to workers assigned earlier tokens to remain idle while waiting for workers with later tokens to complete, lowering the GPU utilization almost by half. We address this challenge by introducing a load-balancing schedule that routes the extra attention computation of later tokens to those idle workers (§ 3.2). This optimization yields twice throughput of the unbalanced version as shown in Figure 5.

The second challenge is the prohibitive communication overhead. When tokens are distributed to multiple machines, these machines need to communicate key-value tensors and softmax statistics to jointly compute the global attention. The communication volume is nontrivial, leading to large communication overhead, which grows with the context length. By leveraging the attention dependencies, we propose a scheduling technique that overlaps communication and computation by pre-fetching tensors. This successfully hides communication overhead inside the computation time, resulting in a 1.32× end-to-end speedup (Figure 5) compared to a non-overlapping version.

The third challenge is the high computation overhead due to the re-computation in gradient checkpointing (Chen et al., 2016). Gradient checkpointing effectively trades computation for memory by selectively storing intermediate activations (e.g., the inputs of every layer) and recomputing on-the-fly during the backward pass. It has become a standard technique in the training of long-context LLMs to accommodate the prohibitive activation memory (Zheng et al., 2023). However, the recomputation of the FlashAttention causes a high computation overhead in long sequences where the attention dominates the computation time. In § 3.3, we show the recomputation of FlashAttention is unnecessary for its backward pass and propose a novel gradient checkpointing strategy to avoid it. Our new strategy results in an 1.31× speedup (§ 4.6) without introducing any numerical difference.

Our main contributions are:

1. We develop DISTFLASHATTN, a distributed, memory-efficient, exact attention mechanism with sequence parallelism. We propose new optimization techniques to balance the causal computation workloads and overlap computation and computation

to increase GPU utilization and reduce communication overhead for training long-context LLMs. We also propose a rematerialization-aware gradient checkpointing strategy that eliminates redundant forward recomputation of FlashAttention.

2. We perform comprehensive evaluation of DISTFLASHATTN on LLaMA models, against four strong distributed systems. DISTFLASHATTN supports 8× longer sequences with 5.64× compared to Ring Self-Attention, 2 – 8× longer sequences with 1.24 – 2.01× speedup compared to Megatron-LM. DISTFLASHATTN achieves 1.67× and 1.26 – 1.88× speedup compared to Ring Attention and DeepSpeed-Ulysses.

## 2 Related work

**Memory-efficient attention.** Dao et al. (2022) and Lefaudeux et al. (2022) propose to use an online normalizer (Milakov & Gimelshein, 2018) to compute the attention in a blockwise and memory-efficient way. It reduces peak memory usage by not materializing large intermediate states, e.g. the attention softmax matrix. In addition, research on sparse attention computes only a sparse subset of the attention score, which also reduces the memory footprints yet may lead to inferior performance (Beltagy et al., 2020; Sun et al., 2022; Zaheer et al., 2020). In this work, we limit our scope to exact attention.

**Sequence parallelism and ring attention** Ring Self-Attention (Li et al., 2021) is among the first to parallelize Transformers in the sequence dimension. However, its distributed attention design is not optimized for causal language modeling and incompatible with memory-efficient attention, which are crucial for long-context LLM training. Ring Attention (Liu et al., 2023) proposes to compute distributed attention in a memory-efficient blockwise pattern. However, it is also not optimized for causal language modeling, leading to 2× extra computation. DISTFLASHATTN optimizes for both memory-efficient attention and causal language modeling. More recently, DeepSpeed Ulysses (Jacobs et al., 2023) proposes a hybrid parallelism strategy. It computes distributed attention in the tensor model parallelism to address these two problems and utilizes sequence parallelism elsewhere (Shoeybi et al., 2019). We provide head-to-head comparison in Table 4.

**Model Parallelism and FSDP** Tensor Model parallelism (Korthikanti et al., 2023) partitions model parameters and also distributes the activation in parallel LLM training. Pipeline model parallelism (Huang et al., 2019) also partitions the activations. However, it applies high memory pressure to the first pipeline stage. We show in § 4.3 that this leads to a less effective support for long sequences. Thus, we focus on comparing with tensor model parallelism and only consider pipeline parallelism when the number of heads is insufficient for tensor parallelism. Fully sharded data-parallelism (FSDP) (Zhao et al., 2023; Rajbhandari et al., 2020) distributes optimizer states, gradients, and model parameters onto different devices and gathers them on-the-fly. Our work focuses on reducing the activation memory that dominates in long-context training. Therefore, FSDP is orthogonal to our work.

**Gradient checkpointing.** Gradient checkpointing (Chen et al., 2016) trades computation for memory by not storing activations for certain layers and recomputing them during the forward pass. Selective checkpointing (Korthikanti et al., 2023) suggests recomputing only the attention module, as it requires significant memory but relatively few FLOPs (in contexts of smaller length). Checkmate (Jain et al., 2020) finds optimal checkpointing positions using integer linear programming. However, none of these designs have considered the effects of memory-efficient attention kernels, which perform recomputation within the computational kernel to avoid materializing large tensors. In this paper, we demonstrate that by simply altering the checkpointing positions, we can avoid the recomputation of these kernels without introducing any numerical difference.

## 3 Method

In this section, we first present a distributed memory-efficient attention mechanism that distributes the computation along the sequence dimension, DISTFLASHATTN (§ 3.1) in its vanilla form. We then introduce two novel optimizations in DISTFLASHATTN: a load-balanced scheduling strategy for causal language modeling to reduce the computation

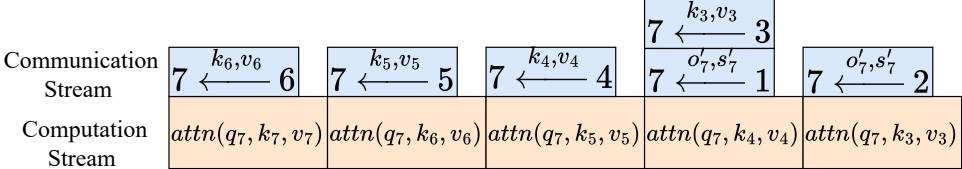

Figure 2: Overlap example in the forward pass of worker 7 out of an 8 worker scnerio. For simplicity, "worker p" is denoted as p.

bubble and an asynchronous communication design that overlaps the communication into computation (§ 3.2). Finally, we propose a new rematerialization-aware checkpointing strategy (§ 3.3) which effectively cuts off the recomputation time in gradient checkpointing when using DISTFLASHATTN in long-context training.

## 3.1 DISTFLASHATTN: distributed memory-efficient attention via sequence parallelism

The goal of DISTFLASHATTN is twofold: (1) distribute a single sequence into multiple workers so they jointly utilize the memory to support a long sequence training; (2) maintain the IO-aware benefits of memory-efficient attention so that training is fast and incurs less memory footprint. In particular, we choose FlashAttention (Dao, 2023) as the paradigm.

---

**Algorithm 1** (Vanilla) DISTFLASHATTN of worker $p$

---

**Require:** $\mathbf{q}_p, \mathbf{k}_p, \mathbf{v}_p$
1: Initialize $\mathbf{o}_p = \mathbf{o}^0, \mathbf{s}_p = \mathbf{s}^0 = [\mathbf{m}^0, \mathbf{l}^0]$, where $\mathbf{o}^0 = \mathbf{0}, \mathbf{l}^0 = \mathbf{0}$, and $\mathbf{m}^0 = [-\infty \cdots -\infty]^T$
2: $\mathbf{o}_p, \mathbf{s}_p = attn(\mathbf{q}_p, \mathbf{k}_p, \mathbf{v}_p, \mathbf{o}_p, \mathbf{s}_p)$
3: **for** $1 \le t < p$ **do**
4:    $\mathrm{r} = (p - t) \pmod{P}$
5:    Fetch from remote: worker p $\xleftarrow{\mathbf{k}_r, \mathbf{v}_r}$ worker r
6:    $\mathbf{o}_p, \mathbf{s}_p = attn(\mathbf{q}_p, \mathbf{k}_r, \mathbf{v}_r, \mathbf{o}_p, \mathbf{s}_p)$
7: **end for**
8: Return $\mathbf{o}_p$.

---

**To distribute the long sequence.** DISTFLASHATTN splits the input sequence consisting of $N$ tokens evenly across $P$ workers (e.g. GPUs) along the sequence dimension. Each worker computes and stores the activations of only a subsequence of $N/P$ tokens. Therefore, it supports training $P\times$ longer with $P$ workers than a single-worker FlashAttention.

Formally, let $\mathbf{q}_p, \mathbf{k}_p, \mathbf{v}_p \in \mathbf{R}^{\frac{N}{P} \times d}$ be the query, key and value of the subsequence on the $p$-th worker ($p = \{1, \cdots, P\}$), where $d$ is the hidden dimension. Considering the most prevalent causal attention in LLMs, worker p computes the attention output $\mathbf{o}_p$ associated with $\mathbf{q}_p$:

$$\mathbf{o}_p = \text{Softmax}(\frac{\mathbf{q}_p[\mathbf{k}_1, ..., \mathbf{k}_p]^T}{\sqrt{d}})[\mathbf{v}_1, ..., \mathbf{v}_p] \tag{1}$$

**To maintain the IO-awareness**. Naïvely, each worker could gather all the keys and values associated with other subsequences and then locally computes $\mathbf{o}_p$ by invoking the existing single machine FlashAttention. However, this gathering introduces memory pressure by having to store the full list of keys and values locally, a total size of $\mathbf{R}^{2N \times d}$.

Fortunately, the block-wise nature of the single-worker FlashAttention only requires **one** block of keys and values in each iteration of its algorithm, Leveraging this observation, we

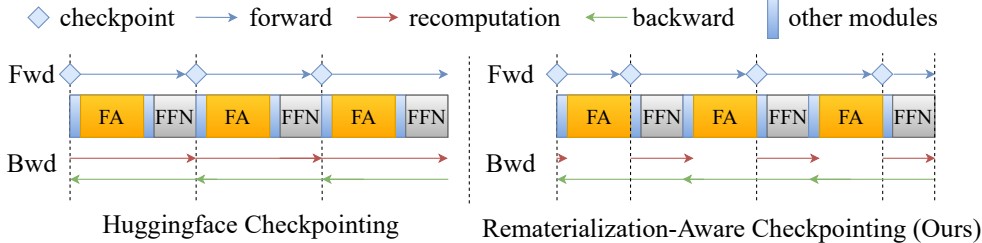

Figure 3: Comparison of HuggingFace gradient checkpointing strategy and our materialization-aware gradient checkpointing strategy. Note that our checkpointing strategy **saves an entire flash attention forward per layer in recomputation** by simply shifting the checkpoint boundaries without introducing any numerical difference. The checkpointed tensors, i.e., the outputs of FlashAttention(FA), are saved not only for the recomputation of subsequent layers but also the backward computation of the preceding FlashAttention.

compute $\mathbf{o}_p$ iteratively: in each iteration when $r \neq p$, worker p fetches only *one* $\mathbf{k}_r, \mathbf{v}_r$ from a remote worker $r$, It then computes partial attention results based on $\mathbf{q}_p$ and $\mathbf{k}_r, \mathbf{v}_r$ and perform proper rescaling by invoking the single-worker FlashAttention kernel. To perform proper rescaling between iterations, each worker also needs to maintain a copy of softmax statistics[1] $\mathbf{s}_p \in \mathbf{R}^{\frac{2N}{P}}$. Computing in this iterative manner, each worker also stores the key and value of one subsequence of size $\mathbf{R}^{\frac{2N \times d}{P}}$, a factor of $\frac{1}{P}$ memory of the naïve design. We refer to Dao et al. (2022) for more details of the single-worker FlashAttention. We denote each iteration of the partial attention result and the rescaling as $attn(\mathbf{q}_p, \mathbf{k}_r, \mathbf{v}_r, \mathbf{s}_p)$, and present the vanilla DISTFLASHATTN algorithm in Algorithm 1. In Appendix A, we show how to implement the $attn(\cdot)$ kernel from Dao (2023) in pseudo-code.

## 3.2 Load balanced scheduling with communication and computation overlap

**Load-balanced scheduling.** In causal attention, each token only attends to its previous tokens, i.e. the p-th worker computes $attn(\mathbf{q}_p, \mathbf{k}_r, \mathbf{v}_r)$ for all $r \leq p$. This introduces a workload imbalance between workers: a worker with a larger $p$ computes more $attn(\cdot)$ (Figure 1 (a)). Using the scheduling described in § 3.1, the idle fraction is $\frac{P^2-P}{2P^2}$ ($\rightarrow \frac{1}{2}$ when $P \rightarrow \infty$), which means roughly half of the workers are idle. To reduce this idle time, we let worker $r_1$ that has finished all its $attn(\cdot)$ computations (i.e., the "helper") perform attention computation for worker $r_2$ with heavier workload, as shown in Figure 1 (b).

Notably, the "helper" $r_1$ needs to communicate softmax statistics and the partial attention output to the original worker $r_2$, so that worker $r_2$ can update its local copy of statistics and output correctly (Algorithm 2). This update function is denoted as $rescale(\cdot)$ and updates the partial output and statistics in the same way as how Dao (2023) updates results from two block execution. This scheduling gives an average idle time fraction:

$$X = \begin{cases} 0, & \text{P is odd} \\ \frac{1}{2P}, & \text{P is even} \end{cases} \qquad (2)$$

Note that when P is even, the idle time is asymptotically 0 to more workers. We provide an illustration with 8 workers in Figure 1. While we focus on the exact attention mechanism, we also discuss sparse patterns in Appendix D.

**Communication and computation overlap.** DISTFLASHATTN relies on peer-to-peer (P2P) communication to fetch $\mathbf{k}_r, \mathbf{v}_r$ (or $\mathbf{q}_r$ in the load-balanced scheduling) from remote workers before computing $attn(\cdot)$. However, these communications can be naturally overlapped. To simplify the equations, we use the unbalanced schedule to describe the intuition, while the

---

[1]These are statistics $l$ and $m$ in FlashAttention words.

final DISTFLASHATTN implementation are equipped with both optimizations. Precisely, these two operations are parallelized:

$$\text{Fetch : worker } p \xleftarrow{\mathbf{k}_{r+1},\mathbf{v}_{r+1}} \text{worker } r+1$$
$$\text{Compute : } attn(\mathbf{q}_p, \mathbf{k}_r, \mathbf{v}_r, \mathbf{s}_p) \tag{3}$$

Thus, in the next iteration, $\mathbf{k}_{r+1}, \mathbf{v}_{r+1}$ are already stored in the memory of worker p, without blocking the next iteration's computation. In modern accelerators, this can be done by placing the attention computation kernel in the main GPU stream, and the P2P communication kernel in another stream, where they can run in parallel (Zhao et al., 2023). We demonstrate the overlapped scheduling for worker 7 in the 8-worker scenario in Figure. 2. Empirically, we find this optimization effectively reduces the communication overhead by hiding the communication time inside the computation time (§ 4.6).

### 3.3 Rematerialization-aware checkpointing strategy

Gradient checkpointing (Chen et al., 2016) is a de-facto way of training long-context transformers. Often, the system uses heuristics to insert gradient checkpoints at the boundary of each Transformer layer (Wolf et al., 2019). However, with the presence of Dao et al. (2022), we find the previous gradient checkpointing strategy causes a redundant recomputation of the FlashAttention forward kernel. Precisely, when computing the gradient of the MLP layer, Wolf et al. (2019) re-computes the forward of the entire Transformer layer including FlashAttention. During this process, the FlashAttention backward kernel re-computes the softmax block-wisely again to reduce memory usage. Essentially, this is because FlashAttention does not materialize the intermediate values during the forward, and recomputes it during the backward, regardless of the re-computation in the outer system level (e.g., the HuggingFace gradient checkpointing (Wolf et al., 2019)).

To tackle this, we propose to insert checkpoints at the output of the FlashAttention kernel, instead of at the Transformer layer boundary. We use each checkpoint not only for the recomputation of its subsequent modules but also for the backward computation of its preceding FlashAttention module without recomputation. Thus we only need to compute the forward of FlashAttention once, effectively avoiding all recomputations of FlashAttention as shown in Figure 3.

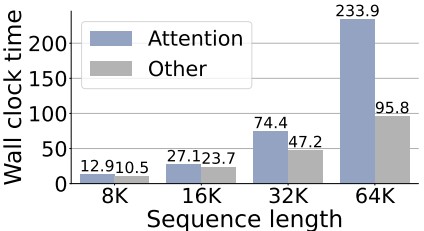

Figure 4 shows that attention dominates in the forward pass with in long sequences, which indicates our method saves $\sim 0.23 \times 32$ (i.e., $\sim 7$) seconds when training a 64K sequence example on Llama-7b on a single machine. In addition, this saves a communication brought by our DISTFLASHATTN forward in

Figure 4: Time breakdown of attention versus other modules in a forward pass, measured with Flash-Attention (Dao, 2023) on a single 40GB A100 GPU. (Unit: ms)

the distributed training scenario. We benchmark the end-to-end speedup brought by this materialization-aware checkpointing strategy in § 4.6.

## 4 Experiments

We evaluate DISTFLASHATTN together with our new checkpointing strategy against alternative distributed approaches for long-context LLMs training. Our primary baseline is Megatron-LM (Shoeybi et al., 2019), used in tandem with FlashAttention, which serves as a robust baseline extensively adopted within the industry. In Appendix C, we also show a theoretical analysis on its high communication volume. We also provide a comparison with the previous sequence-parallel system (Li et al., 2021). In addition, we include comparison to recent systems including DeepSpeed-Ulysses and Ring Attention (Jacobs et al., 2023; Liu et al., 2023). In the ablation study, we delineate the individual contributions of each component of our methodology, specifically load balancing, computation-communication overlapping, and rematerialization-aware checkpointing, towards the overall performance enhancement. Code implementation details can be found in Appendix 4.1.

Table 1: Per iteration wall-clock time of DISTFLASHATTN and Megatron-LM (Korthikanti et al., 2023) (Unit: seconds). Speedup in bold denotes the better of the two systems in the same configuration. Time measured with 2 DGX boxes.

| Method | # GPUs | Sequence Length | | LLaMA-7B | | LLaMA-GQA | | LLaMA-33H | |
| | | Per GPU | Total | Time | speedup | Time | speedup | Time | speedup |
|---|---|---|---|---|---|---|---|---|---|
| Megatron-LM | 1x8 | 8K | 64K | 6.81 | 1.0x | 6.60 | 1.0x | 8.37 | 1.0x |
| | 1x8 | 16K | 128K | 20.93 | 1.0x | 20.53 | 1.0x | 25.75 | 1.0x |
| | 1x8 | 32K | 256K | 72.75 | 1.0x | 71.93 | 1.0x | 90.21 | 1.0x |
| DISTFLASHATTN | 1x8 | 8K | 64K | 5.98 | **1.14x** | 5.61 | **1.18x** | 6.08 | **1.38x** |
| | 1x8 | 16K | 128K | 17.26 | **1.21x** | 16.86 | **1.22x** | 17.77 | **1.45x** |
| | 1x8 | 32K | 256K | 58.46 | **1.24x** | 57.01 | **1.26x** | 59.96 | **1.50x** |
| Megatron-LM | 2x8 | 8K | 128K | 14.26 | 1.0x | 14.21 | 1.0x | 20.63 | 1.0x |
| | 2x8 | 16K | 256K | 43.44 | 1.0x | 43.20 | 1.0x | 62.78 | 1.0x |
| | 2x8 | 32K | 512K | 147.06 | 1.0x | 146.38 | 1.0x | 216.70 | 1.0x |
| DISTFLASHATTN | 2x8 | 8K | 128K | 12.75 | **1.12x** | 9.74 | **1.46x** | 13.12 | **1.57x** |
| | 2x8 | 16K | 256K | 30.21 | **1.44x** | 28.49 | **1.52x** | 31.33 | **2.00x** |
| | 2x8 | 32K | 512K | 106.37 | **1.38x** | 102.34 | **1.43x** | 107.76 | **2.01x** |

**Cluster setup.** We evaluate our method and the baselines in (1) A single A100 DGX box with 8x80 GB GPUs. These GPUs are connected with NVLink; (2) 2 DGX boxes with the same setting. These two boxes are interconnected by a stable 100 Gbps Infiniband. This is a representative setting for cross-node training, where the communication overhead is large. Unless otherwise stated, this is our default setup. (3) Our in-house development cluster with 2x8 A100 40GB GPUs. This cluster has unstable inter-node bandwidth. Due to the limited computational budget, we report some peripheral results on this cluster.

**Model setup.** We evaluate our system on LLaMA-7B and its variants, encompassing four sets of model architectures in total: two with regular attention heads and two with irregular ones. We note both categories are important in real-world applications.

**With regular attention heads.** (1) multi-head attention (MHA) models: LLaMA-7B with 4096 hidden size and 32 self-attention heads (Touvron et al., 2023); (2) grouped-query attention(GQA) models: LLaMA-GQA (Ainslie et al., 2023), same as LLaMA-7B but with 8 key-value heads, each shared by 4 queries as a group. During attention computation, it will first replicate to 32 heads to perform matrix multiplication with the correct shape.

**With irregular attention heads.** In addition, we benchmark the following variants that have appeared in applications but have not received much attention regarding their system efficiency: (3) models with an irregular (e.g., non-power-of-two) number of attention heads[2]: We intentionally test our systems and baselines on LLaMA-33H, which has the same configuration as LLaMA-7B but with 33 normal self-attention heads per layer. (4) models with fewer attention heads[3]: According to the recipe in Liu et al. (2021), we designed LLaMA-16H, LLaMA-8H, LLaMA-4H, and LLaMA-2H with 16, 8, 4, and 2 heads, respectively, as a proof of concept for situations when the number of heads is insufficient to further scale up model parallelism with limited resources. We keep the number of attention heads by scaling the number of layers properly[4] and keep the intermediate FFN layer size the same to make the model sizes still comparable. For example, LLaMA-16H has 16 attention heads per layer, a hidden size of 2048, an FFN layer of size 11008, and 64 layers.

### 4.1 Implementation Details

We build the kernel of DISTFLASHATTN, modifying from the Triton kernel of FlashAttention2 in 500 lines of codes (LoCs). We implement the load balancing and overlapping

---

[2]For example, GPT-2-XL has 25 attention heads, GPT-2 has 12 attention heads, LLaMA-33B and its fine-tuned versions (e.g., Tulu-30B) have 52 attention heads, Whisper-large has 20 attention heads, and Falcon-7B has 71 attention heads (Radford et al., 2019; Almazrouei et al., 2023; Ivison et al., 2023).

[3] Liu et al. (2021) finds fewer attention heads with more layers increase the performance.

[4]For instance, LLaMA-7B has 32 attention heads and 32 layers, thus LLaMA-16H has 16 attention heads per layers and 64 layers.

scheduling n Python and NCCL Pytorch bindings in 1000 LoCs (Paszke et al., 2019; Jeaugey, 2017), and the checkpointing strategy in 600 lines of Pytorch. We use block sizes of 128 and the number of stages to 1 in the kernel for the best performance in our cluster. We evaluate DISTFLASHATTN using FSDP (inter-node if applicable) so that it consumes similar memory than the Megatron-LM baseline for a fair comparison (Zhao et al., 2023). For fair comparisons, we run all comparisons using the same attention backend. We also add support for Megatron-LM so that comparing with them can produce a more insightful analysis: (1) not materializing the causal attention mask, greatly reducing the memory footprint. For instance, without this support, Megatron-LM will run out of memory with LLaMA-7B at a sequence length of 16K per GPU. (2) head padding where the attention heads cannot be divided by device number. All results are gathered with Adam optimizer, 10 iterations of warm-up, and averaged over the additional 10 iterations.

## 4.2 Comparison with Megatron-LM on MHA and GQA models

**Multi-head attention (MHA).** On the LLaMA-7B model (Table 1), our method achieves **1.24×** and **1.44×** speedup compared to Megatron-LM in single-node and cross-node setting, up to the longest sequence length we experiment. This is a joint result of our overlapping communication technique and our rematerialization-aware checkpointing strategy. We analyze how much each factor contributes to this result in the ablation study (§ 4.6).

Table 2: The maximal sequence length Per GPU supported by DISTFLASHATTN and Megatron-LM with tensor parallelism and pipeline parallelism on 16xA100 40GB GPUs.

|  | 16H | 8H | 4H | 2H |
|---|---|---|---|---|
| Megatron TP+DP | 512K | 256K | 128K | 64K |
| Megatron TP+PP | 512K | 256K | 256K | 128K |
| DISTFLASHATTN | 512K | 512K | 512K | 512K |

**Grouped-query attention (GQA).** On GQA model, DISTFLASHATTN communicates less volume due to the reduction of size of keys and values. On the contrary, the communication of Megatron-LM remains the same because it does not communicate keys and values. Thus, DISTFLASHATTN achieves a higher speedup on LLaMA-GQA model (Table 1).

## 4.3 Comparison with Megatron-LM on models with irregular or less number of heads

**In support of irregular numbers of heads.** Megatron-LM assumes the number of attention heads is divisible by the model parallelism degree. For example, it supports parallelism degrees of 2, 4, 8, 16, and 32 for models with 32 attention heads. However, it needs to pad dummy heads when the number of heads is not divisible by the ideal parallelism degree. For example, it needs to pad 15 dummy heads to support a parallelism degree of 16 for models with 33 attention heads (e.g., LlmaMA-33H), leading to a substantial computation wastage of 45.5%. As shown in Table 1, we observe a **1.50×** and **2.01×** speedup (an additional 20% and 45% speedup compared to LLaMA-7B cases, aligned with the theoretical analysis).

**In support of less number of heads.** When the number of GPUs exceeds the number of attention heads, Megatron-LM allows three possible solutions: (1) Pad dummy heads as in the LLaMA-33H scenario. However, the percentage of dummy heads almost directly translates to the percentage of slowdown in long sequences where attention computation dominates. (2) Use data parallelism for excess GPUs. However, data parallelism does not reduce per sequence memory usage, and thus can not jointly support longer sequences. (3) Use pipeline parallelism. However, the memory usage at each stage of the pipeline is not evenly distributed, limiting the maximal sequence length supported. For instance, in the LLaMA-2H experiment, we find that different stages consume from 18GB to 32GB in a 64K sequence length (Section B). In addition, using pipeline parallelism introduces an extra fraction of GPU idle time. We demonstrate the effect of using the latter two solutions in Table 2. In 16 A100 40GB GPUs, DISTFLASHATTN supports 2× and 8× longer sequences.

## 4.4 Comparison with Ring Self-Attention (RSA) and Ring Attention

Ring self-attention (RSA) (Li et al., 2021) communicates tensors in a ring fashion. We first report the maximal sequence length of RSA and DISTFLASHATTN in Table 3, and found that DISTFLASHATTN supports at least 8x longer sequences than RSA. This is mainly because RSA is not natively compatible with memory-efficient attention. We further measure the iteration time with the maximal sequence length that RSA can support in Table 3, and find that DISTFLASHATTN is 4.45x - 5.64x faster than RSA. This speedup includes a 2x improvement from our causal workload balancing optimization and additional gains from the overlapping optimization and extending memory-efficient attention to the distributed setting. Both experiments are conducted with the Llama-7B model and on the DGX cluster.

Table 3: Max sequence length and per iteration time (seconds) compared with RSA.

| | 1 Node | 2 Nodes |
|---|---|---|
| RSA | 32K | 64K |
| DISTFLASHATTN | > 256K | > 512K |

| | 1 Node (32K) | 2 Nodes (64K) |
|---|---|---|
| RSA | 14.10 | 30.49 |
| DISTFLASHATTN | 2.50 | 6.85 |
| Speedup | 5.64x | 4.45x |

Ring Attention (Liu et al., 2023) implements distributed attention in a memory-efficient manner. The key difference between Ring Attention and DISTFLASHATTN is DISTFLASHATTN has additional optimization of causal workload balancing and a better gradient checkpoint strategy. The implementation of Ring Attention uses a different framework from ours (Jax versus PyTorch). To provide a fair comparison, we consider our ablation version in § 4.6 as a PyTorch implementation of Ring Attention. § 4.6 provides a detailed analysis. In 8-GPU setting, we observe a 1.67× speedup (7.5× versus 4.5× speedup compared to a single GPU FlashAttention) over the design of Ring Attention.

## 4.5 Comparison with DeepSpeed Ulysses

Table 4: Per iteration wall-clock time (seconds) of DISTFLASHATTN and DeepSpeed Ulysses.

| Method | # GPUs | Sequence Length Per GPU | Total | Time | Speedup | # GPUs | Sequence Length Per GPU | Total | Time | Speedup |
|---|---|---|---|---|---|---|---|---|---|---|
| | | Llama-7B | | | | | Llama-33H | | | |
| DeepSpeed-Ulysses | 2x8 | 16K | 256K | 37.53 | 1.0x | 2x8 | 16K | 256K | 56.63 | 1.0x |
| | 2x8 | 32K | 512K | 134.09 | 1.0x | 2x8 | 32K | 512K | 202.89 | 1.0x |
| DISTFLASHATTN | 2x8 | 16K | 256K | 30.21 | **1.21x** | 2x8 | 16K | 256K | 31.33 | **1.81x** |
| | 2x8 | 32K | 512K | 106.37 | **1.26x** | 2x8 | 32K | 512K | 107.76 | **1.88x** |

DeepSpeed-Ulysses (Jacobs et al., 2023) uses all-to-all primitive to reduce the communication. We evaluate a representative subset of experiments in Table 4 due to computational budget limit. On experiments with regular heads models (Llama-7B), DISTFLASHATTN achieves 1.26 × speedup. On experiments on irregular heads models (Llama-33H), DISTFLASHATTN achieves 1.88× speedup. Essentially, DeepSpeed-Ulysses also paralleize on the attention head dimension, and suffer from the same problems as analyzed in § 4.3.

## 4.6 Ablation Study

**Effect of Load Balancing** We study load balancing on an attention forward pass of LLaMA-7B model, on 8 A100 40GB GPUs (Figure 5). The backward pass follows a similar analysis. With an unbalanced schedule (Figure 1), the total work done is 36, where the total work could be done in 8 units of time is 64. Thus, the expected speedup is 4.5x. In the balanced schedule, the expected speedup is 7.2x. We scale the total sequence length from 4K to 256K. The unbalanced version saturates in 4.5x speedup compared to a single GPU FlashAttention, while the balanced version saturates at 7.5× speedup. Both of them align with our theoretical analysis and show the effectiveness of the balanced scheduling.

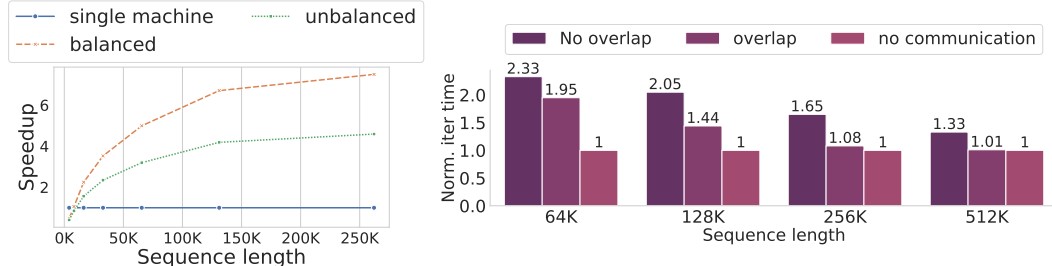

Figure 5: Effect of balanced schedule (left) and the effect of overlapping (right).

**Effect of overlapping communication and computation.** We study the overlapping communication on LLaMA-7B and 2 DGX boxes (Figure 5). We find that overlapping greatly reduces the communication overhead. On a global sequence length of 128K, the communication overhead is reduced from 105% to 44%. This overlapping scheme maximizes its functionality when the communication overhead is less than 100%, where all communication can be potentially overlapped. Empirically, we find the system only exhibits 8% and 1% overhead in these cases, a close performance to an ideal system without communication.

**Effect of rematerialization-aware checkpointing.** We show in Table 5 the effects of the proposed rematerialization-aware gradient checkpointing. Our method achieves 1.16x, 1.24x, and 1.31x speedup at the sequence length of 8K, 16K, and 32K per GPU respectively. The materialization-aware checkpointing strategy speeds up more at longer sequence lengths where the attention dominates the computation.

### 4.7 Partition on the attention heads or sequence dimension

Megatron-LM and DeepSpeed-Ulysses are distributed systems that partition on attention heads. While it allows seamless integration with the FlashAttention kernel, it has certain limitations. These includes: (1) Not being able to utilize the pattern inside the attention module, missing opportunities to reduce communication for causal, and grouped-query attention (See § C). (2) not flexible to support arbitrary number of attention heads, and (3) Importantly, its scala-

Table 5: Our checkpointing algorithm ("Our ckpt") versus HuggingFace strategy ("HF ckpt") on 8 A100s (batch size 1, Unit: seconds).

| Method | Sequence Length Per GPU | | | | | |
|---|---|---|---|---|---|---|
| | 1K | 2K | 4K | 8K | 16K | 32K |
| HF ckpt | 0.84 | 1.29 | 2.64 | 6.93 | 21.44 | 76.38 |
| Our ckpt | 0.84 | 1.36 | 2.50 | 5.98 | 17.26 | 58.46 |
| Speedup | 1.0x | 0.94x | **1.06x** | **1.16x** | **1.24x** | **1.31x** |

bility is limited by the number of attention heads (in the scale of several to several dozens), while the maximal number of parallelism degree for sequence parallelism is at least several thousands. Given these reasons, we think it is worth pursuing the sequence parallelism paradigm when distributing the attention module.

## 5 Conclusion

In this work, we introduce DISTFLASHATTN, a distributed memory-efficient attention prototype for long-context transformer training based on sequence parallelism. DISTFLASHATTN presents novel system optimizations including load balancing for causal language modelings, overlapped communication with computation in the distributed attention computation, and a re-materialization-aware checkpointing strategy. Experiments evaluate multiple families of transformer models and on different cluster types, and over four strong distributed system baselines. In particular, DISTFLASHATTN has demonstrated up to 2.01× speedup and scales up to 8x longer sequences, compared to the popular system, Megatron-LM with FlashAttention.

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

## A From FlashAttention to $attn(\cdot)$ in DISTFLASHATTN

In this section, we provide the details of the $(attn)(\cdot)$ kernel in DISTFLASHATTN.(Alg 3). For conceptual simplicity, we demonstrate it in the most vanilla version, without the actual scheduling (e.g. load balancing and overlapping). We also demonstrate it with the causal language modeling objective. The standalone attention is mainly borrowed from the FlashAttention2 paper (Dao, 2023). To make it compatible with DISTFLASHATTN, we mainly revised the several points:

1. Accumulate results statistics $o$, $m$ and $l$ from previous computation, instead of initializing them inside the function.

2. Pass an extra argument "last", which means whether this is the last chunk of attention computation. Only when it is true, we compute the logsumexp $L$.

At a high level, on a worker $p$, DISTFLASHATTN first initializes local statistics $o, m, l, L$. Then DISTFLASHATTN loops over all its previous workers. In each iteration, it fetches the key and the value from a worker and invokes the revised standalone attention to update local statistics. At the end of the iteration, it needs to delete the remote key and value from HBM so that the memory does not accumulate. At the last iteration of the loop, it additionally calculates the logsumexp according to the final $m$ and $l$ (triggered by the "last" variable in the algorithm). At the end of the forward pass, worker $p$ has the correct $m, l, L$. The backward pass is similar and conceptually simpler because we do not need to keep track of statistics such as $m$ and $l$. Instead, we only need to use the logsumexp stored in the forward pass.

---

**Algorithm 2** (Balanced) DISTFLASHATTN of worker $p$

---

**Require:** $\mathbf{q}_p, \mathbf{k}_p, \mathbf{v}_p$
 1: Initialize $\mathbf{o}_p = \mathbf{o}^0$ , $\mathbf{s}_p = \mathbf{s}^0 = [\mathbf{m}^0, \mathbf{l}^0]$, where $\mathbf{o}^0 = \mathbf{0}$, $\mathbf{l}^0 = \mathbf{0}$, and $\mathbf{m}^0 = [-\infty \cdots - \infty]^T$
 2: $\mathbf{o}_p, \mathbf{s}_p = attn(\mathbf{q}_p, \mathbf{k}_p, \mathbf{v}_p, \mathbf{o}_p, \mathbf{s}_p)$
 3: **for** $1 \leq \mathrm{t} \leq \lfloor \frac{P}{2} \rfloor$ **do**
 4:     $r = (p - t) \pmod{P}$
 5:     **if** $p > \mathrm{t}$ **then**
 6:         Fetch key, value from remote: $p \xleftarrow{\mathbf{k}_t, \mathbf{v}_t} r$
 7:         $\mathbf{o}_p, \mathbf{s}_p = attn(\mathbf{q}_p, \mathbf{k}_r, \mathbf{v}_r, \mathbf{o}_p, \mathbf{s}_p)$
 8:         **if** $\mathrm{t} \neq \lfloor \frac{P}{2} \rfloor$ **and** $(p + t) > \mathrm{P}$ **then**
 9:             $r_2 = (p + t) \pmod{P}$
10:             Fetch result from remote: $p \xleftarrow{\mathbf{o}'_p, \mathbf{s}'_p} r_2$
11:             $\mathbf{o}_p, \mathbf{s}_p = rescale(\mathbf{o}_p, \mathbf{s}_p, \mathbf{o}'_p, \mathbf{s}'_p)$
12:         **end if**
13:     **else**
14:         **if** $\mathrm{t} \neq \lfloor \frac{P}{2} \rfloor$ **then**
15:             Fetch query from remote: $p \xleftarrow{\mathbf{q}_r} r$
16:             $\mathbf{o}_r, \mathbf{s}_r = attn(\mathbf{q}_r, \mathbf{k}_p, \mathbf{v}_p, \mathbf{o}^0, \mathbf{s}^0)$
17:             Send result to remote: $p \xrightarrow{\mathbf{o}_r, \mathbf{l}_r, \mathbf{m}_r} r$
18:         **end if**
19:     **end if**
20: **end for**
21: Return $\mathbf{o}_p$.

---

## B Memory Consumption for Pipeline Parallelism

In this section, we show the memory consumption of Megatron-LM when training with tensor parallelism and pipeline parallelism. As presented in table 6, memory consumption

---

**Algorithm 3** DISTFLASHATTN Pseudo code (forward pass)

---

**Require:** Matrices $\mathbf{Q}^p, \mathbf{K}^p, \mathbf{V}^p \in \mathbb{R}^{\frac{N}{\mathbb{P}} \times d}$ in HBM, block sizes $B_c$, $B_r$, rank
    **function** standalone_fwdq, k, v, o, $\ell$, m, causal, last

1: Divide $q$ into $T_r = \left\lceil \frac{N}{\mathbb{P}B_r} \right\rceil$ blocks $q_1, \ldots, q_{T_r}$ of size $B_r \times d$ each,

2: and divide $k, v$ in to $T_c = \left\lceil \frac{N}{\mathbb{P}B_c} \right\rceil$ blocks $k_1, \ldots, k_{T_c}$ and $v_1, \ldots, v_{T_c}$, of size $B_c \times d$ each.

3: Divide the output $o \in \mathbb{R}^{\frac{N}{\mathbb{P}} \times d}$ into $T_r$ blocks $o_i, \ldots, o_{T_r}$ of size $B_r \times d$ each, and divide the
    logsumexp $L$ into $T_r$ blocks $L_i, \ldots, L_{T_r}$ of size $B_r$ each.

4: **for** $1 \leq i \leq T_r$ **do**

5:     Load $q_i$ from HBM to on-chip SRAM.

6:     Load $o_i \in \mathbb{R}^{B_r \times d}, \ell_i \in \mathbb{R}^{B_r}, m_i \in \mathbb{R}^{B_r}$ from HBM to on-chip SRAM as $o_i^{(0)}, \ell_i^{(0)}, m_i^{(0)}$.

7:     **for** $1 \leq j \leq T_c$ **do**

8:       **if** causal and $i \leq j$ **then**

9:         Continue

10:       **end if**

11:       Load $k_j, v_j$ from HBM to on-chip SRAM.

12:       On chip, compute $s_i^{(j)} = q_i k_j^T \in \mathbb{R}^{B_r \times B_c}$.

13:       On chip, compute $m_i^{(j)} = \max(m_i^{(j-1)}, \text{rowmax}(s_i^{(j)})) \in \mathbb{R}^{B_r}$, $\tilde{p}_i^{(j)} = \exp(S_i^{(j)} - m_i^{(j)}) \in \mathbb{R}^{B_r \times B_c}$ (pointwise), $\ell_i^{(j)} = e^{m_i^{j-1} - m_i^{(j)}} \ell_i^{(j-1)} + \text{rowsum}(\tilde{p}_i^{(j)}) \in \mathbb{R}^{B_r}$.

14:       On chip, compute $o_i^{(j)} = \text{diag}(e^{m_i^{(j-1)} - m_i^{(j)}})^{-1} o_i^{(j-1)} + \tilde{p}_i^{(j)} v_j^p$.

15:     **end for**

16:     On chip, compute $o_i = \text{diag}(\ell_i^{(T_c)})^{-1} o_i^{(T_c)}$.

17:     Write $o_i$ to HBM as the $i$-th block of $o$.

18:     **if** last **then**

19:       On chip, compute $L_i = m_i^{(T_c)} + \log(\ell_i^{(T_c)})$.

20:       Write $L_i$ to HBM as the $i$-th block of $L$.

21:     **end if**

22: **end for**

23: Return $o, \ell, m$ and the logsumexp $L$.
    **end function**

24: Initialize $\mathbf{O}^p = (0)_{\frac{N}{\mathbb{P}} \times d} \in \mathbb{R}^{\frac{N}{\mathbb{P}} \times d}, \ell^{(p)} = (0)_{\frac{N}{\mathbb{P}}} \in \mathbb{R}^{\frac{N}{\mathbb{P}}}, m^p = (-\infty)_{\frac{N}{\mathbb{P}}} \in \mathbb{R}^{\frac{N}{\mathbb{P}}}$.

25: $\mathbf{O}^p, \ell^p, m^p, L^p = \text{standalone\_fwd}(\mathbf{Q}^p, \mathbf{K}^p, \mathbf{V}^p, \mathbf{O}^p, \ell^p, m^p, \text{True}, \text{p=1})$

26: **for** $1 \leq r < p$ **do**

27:     Receive $\mathbf{K}^r$ and $\mathbf{V}^r$ from **Remote** worker $r$ into HBM.

28:     $\mathbf{O}^p, \ell^p, m^p, L^p = \text{standalone\_fwd}(\mathbf{Q}^p, \mathbf{K}^y, \mathbf{V}^y, \mathbf{O}^p, \ell^p, m^p, \text{False}, \text{r=(p-1)})$

29:     Delete $\mathbf{K}^r$ and $\mathbf{V}^r$ from HBM.

30: **end for**

31: Return the output $\mathbf{O}^p$ and the logsumexp $L$.

---

Table 6: The memory consumption of Megatron-LM when training Llama-2H with tensor parallelism (degree=2) and pipeline parallelism (degree=8) on 16xA100 40GB GPUs at the sequence length of 128K. The memory consumption is highly uneven across pipeline stages.

|  | Worker 1 | Worker 2 | Worker 3 | Worker 4 | Worker 5 | Worker 6 | Worker 7 | Worker 8 |
|---|---|---|---|---|---|---|---|---|
| node 1 | 31.5GB | 31.4GB | 28.7GB | 28.7GB | 26.0GB | 26.0GB | 24.6GB | 24.6GB |
| node 2 | 21.8GB | 21.8GB | 20.5GB | 20.5GB | 17.9GB | 17.8GB | 32.0GB | 32.1GB |

are uneven across different pipeline stages, making scaling through pipeline parallelism hard.

## C  Communication and memory analysis

Denote the hidden dimension as $d$. In DISTFLASHATTN, every worker needs to fetch key and value chunks both of size $\frac{N}{P}d$ before performing the corresponding chunk-wise computation. Thus, the total communication volume in the $P$-workers system is $2 \times \frac{N}{P}d \times P = 2Nd$. With the causal language objective, half of the keys and values do not need to be attended, halving the forward communication volume to $Nd$. In the backward pass, DISTFLASHATTN needs to communicate keys, values, and their gradients, which has $2Nd$ volume. It adds up to $3Nd$ as the total communication volume for DISTFLASHATTN. In Megatron-LM (Korthikanti et al., 2023), each worker needs to perform six all-gather and four reduce-scatter on a $\frac{N}{P}d$ size tensor, thus giving a total communication volume of $10Nd$. Considering gradient check-pointing, Megatron-LM will perform communication in the forward again, giving a total volume of $14Nd$. On the other hand, our communication volume remains $3Nd$ because of the rematerialization-aware strategy. In conclusion, DISTFLASHATTN achieves 4.7x communication volume reduction compared with Megatron-LM.

In large model training, we usually utilize techniques such as FSDP to also reduce the memory consumed by model weights. In this case, We note that the communication introduced by FSDP is only proportional to the size of model weights, which does not scale up with long sequence length. We show the end-to-end speedup with FSDP in Table 1. For clarity, we also note that DISTFLASHATTN is orthogonal to FSDP and by default can be used by itself. In the situations where the model uses MQA or GQA, DISTFLASHATTN further saves the communication volumes by the shared key and values, which we discuss in detail in § 4.2. However, we also note that this is a theoretical analysis, where the wall-clock time may differ because of factors such as implementations. In the experiment section, we provide wall-clock end-to-end results for comparison.

## D  Discussion on sparse attention

While this paper focuses on discussing the exact attention mechanism, we also provide possible solutions for sparse patterns and hope it can inspire future works. In particular, we discuss load balancing for local sliding windows and global attention (Beltagy et al., 2020).

**Local sliding windows** For local sliding windows, the workload is naturally (near) balanced, regardless of single directional or bidirectional attention. Thus, simply disregarding the attention logic to non-local workers suffices. For instance, in exact attention, worker 7 needs to compute attention to all other workers. If the sliding window has a number of tokens equal to that of one worker, then worker 7 only needs to attend to itself and tokens in worker 6. In other words, it only needs to fetch key and value from worker 6, and compute attention. In terms of implementation change, the system merely needs to change the end condition of the for loop (from looping worker 1 - worker 7 to looping only from worker 6 - worker 7).

**Global attention** In global attention, there are a certain number of global tokens that all later tokens need to attend to, which are used to capture the global information. To adapt DISTFLASHATTN to this, one solution is to keep a replica of all the global tokens in each worker, which is simple and practical as otherwise, the global tokens will need to be all-

gathered at each time step. The other possibility is to also split the global tokens evenly onto all workers and use all-gather upon computation to further reduce the memory requirement.

