# OpenReview forum: "DISTFLASHATTN: Distributed Memory-efficient Attention for Long-context LLMs Training"
_colmweb.org/COLM/2024/Conference — COLM_

### Official Review · Reviewer_XmqB · 2024-05-11

**Rating:** 6
**Confidence:** 3
**Ethics Flag:** 1

**Summary:**

The recently introduced FlashAttention re-packages the computation of attention ($softmax(QK^\top)\cdot V$) so that it reduces the number of I/O operations in a _single_ GPU setup. This led to substantial improvement over the earlier implementations of Attention. However, FlashAttention was only designed for a single GPU case.

This paper considers the problem of extending FlashAttention to work for the case of a cluster of GPUs. Specifically the paper identifies three bottlenecks in this extension:

1. For the case of causal Attention, one would expect to see a factor $2\times$ speedup over non-causal Attention (since the former only performs half the number of FLOPs over the latter). However, if one is not careful in how `chunks' of FlashAttention computations are distributed then lot of processors would sit 'idle.' The idea in the paper is to re-distribute the computation of the sub-blocks across processors.
2. The problem with re-distributing 'chunks' of the Attention computation to various processors is that if not done properly, this could lead to a large communication overhead. Thus, to get towards regaining the $2\times$ improvement mentioned above, this paper proposes a better scheduling of these partial computations to reduce the communication. Further, the paper shows how to parallelize the communication and computation tasks at each processor to gain further improvement.
3. To handle really long context length, the now standard practice is to do _gradient checkpointing_, which basically trades memory for re-computation. However, for longer context lengths this recomputation time starts to become a bottleneck. This paper observes that in FlashAttention backward pass recomputation is not needed.

The paper implements the above into a system it calls DistFlashAttention and compares their implementation to existing distributed Attention implementations including Ring (Self)Attention, Megatron-LM and DeepSpeed-Ulysses. It achieves a minimum speedup of $1.26\times$ and a max of $5.64\times$ speedup over various implementations. The speedups for Ring Self-Attention and Megatron-LM also comes along with $2-8\times$ longer sequence lengths. The paper also presents ablation experiments.

Overall, the paper considers an important problem and proposes new system that achieves speedup over existing systems. However, the current writing leaves certain questions unanswered (please see the Question to Authors section for more). Because of these questions, I am rating this a just above threshold of acceptance. However, if the authors are able to address those questions satisfactorily then this paper _could_ be a clear accept for me.

Detailed Comments for Authors
-----------------------------

Below is a list if minor typos/presentation comments:

* [Pg. 2, line -3] "$5.64\times$ compared"-- "speedup" is missing.
* [Pg. 4, line 2] "which" -> "that"
* [Pg. 5, 2 lines above eq (3)] The final implementation of DistFlashAttention is Algorithm 2?

**Questions To Authors:**

1. How were the three bottlenecks identified?
2. Please clarify which bottlenecks are FlashAttention specific and which ones would apply to any Attention implementation.
3. Please clarify which of the proposed ideas appear in other works (like RingAttention) and which ones are new to this paper.
4. What are the main idea(s) in Algorithm 2?
5. What are the max theoretical gains that one can hope from the ideas presented in this paper?

**Reasons To Accept:**

* The paper considers an important problem
* The paper identifies three issues with an obvious distributed implementation of FlashAttention and then proposes fixes for them.
* The experiments shows that the proposed solutions lead to an overall speedup over existing systems.

**Reasons To Reject:**

* Some parts of the paper could do with more explanation
    * How were the three bottlenecks identified? Are these the three most important bottlenecks? If so, how was this determined?
    * Also the first two bottlenecks are not specific to FlashAttention but to Attention in general? Either way, this should be clarified.
    * Some parts of proposed system (e.g. distributing 'shards' of the Attention computation) are not new to the paper (e.g. the sharding is proposed in RingAttention as well-- from what I understand)
* The paper shunts all discussion of its scheduling ideas to the pseudocode in Algorithm 2. The main part of the paper should at the very least state the main idea(s).
* It would be helpful to put the improvements in the experiments in context if it would be possible to give a theoretical bound for what is the maximum improvement that can be achieved with the three main ideas. For the causal part, it is clear that handling that would give a $2\times$ improvement but I do not get a sense of what is the max improvement once can expect from the other ideas. Also do the improvements from the different ideas multiply? I.e. is there reason to believe that the various ideas are not in 'conflict' with each other? Also stating the max theoretical speedups for the ablation experiments would be useful as well.

As mentioned earlier, none of the above weaknesses are inherent and probably can be fixed by revising the writing.

---

> ### Author Rebuttal · Authors · 2024-05-31
>
> We thank the reviewer for acknowledging our contributions. We would like to answer the questions below:
>
> Q1: How were the bottlenecks identified? Clarify which bottlenecks are FlashAttention specific and which ones would apply to any Attention implementation.
>
> A1: We identify these bottlenecks by checking the GPU utilization at each step. To measure the inefficiency, we treat the GPU utilization in the single-GPU setting as an upper bound and compare our GPU utilization in the idstributed setting with it. Based on this metric, we identify the 3 bottlenecks and designed ways to overcome them. The bottleneck of the imbalance workload and the communication overhead are specific to sequence parallelism, and the inefficient gradient checkpointing is specific to FlashAttention in both single-GPU and multi-GPU settings.
>
> Q2: Clarify which ideas appear in other works and which ones are new in this paper.
>
> A2: The idea of sequence parallelism was first introduced by Ring Self-Attention. The extension of FlashAttention with sequence parallelism is new (which is studied in Ring Attention, which is a concurrent work—we developed DistFlashAttn before it was published). In addition, the 3 bottlenecks we identified and solved are new, including imbalance workload (solved with our scheduling algorithm), communication overhead (hidden by asynchronous computation-communication overlap), and inefficient gradient chekcpointing (improved by our rematerialization-aware gradient checkpointing).
>
> Q3: What are the main ideas in Algorithm 2?
>
> A3: The main idea in Algorithm 2 is the load-balancing scheduling added to the naive version of DistFlashAttn, as illustrated in Figure 1. It reduces the time GPUs spend on waiting for others to finish their computation. We thank the reviewer for pointing out its lack of context in the main paper. We will add more description in the main paper.
>
> Q4: What is the max theoretical gians that one can hope from the ideas presented in this paper?
>
> A4: Each component in our algorithm can bring orthogonal benefits that can be combined. Therefore, the speedup can be multiplied. In summary, we could achieve a 2.4x speedup (2x brought by load-balancing scheduling and 1.2 brought by new gradient chekcpointing)  and a support of infinite sequence length (brought by sequence parallelism).
>
> Due to word count limit, we skipped some details. We would love to provide more detailed answers in the second phrase of rebuttal.

---

> > ### Comment · Reviewer_XmqB · 2024-06-02
> > **The 1.2X speedup from gradient checkpointing**
> >
> > Thanks for your responses.
> >
> > Could y'all please give more details on how the $1.2\times$ speedup for gradient checkpoint was calculated?

---

> > > ### Author Response · Authors · 2024-06-03
> > > **Explanation of the speedup**
> > >
> > > Sure! In short, the traditional gradient checkpointing computes 1 normal forward, 1 recomputation forward, and 1 backward. Because FlashAttention recomputes the attention again in its backward kernel to avoid rematerializing the big softmax matrix (i.e., being I/O-aware), the FlashAttention backward equals one recomputation forward plus 1 normal backward. All these add up to 3 forward and 1 backward. Our idea avoids doing two times of forward recomputation, i.e., reducing the total computation from 3 forward + 1 backward to 2 forward + 1 backward, as illustrated in Figure 3. Since backward computation is 2 times the forward, our theoretical gain is ⅕. We thank the reviewer for asking and would like to provide more details upon request!

---

> > > > ### Comment · Reviewer_XmqB · 2024-06-04
> > > >
> > > > Thanks for the explanation.

---

> > > > > ### Author Response · Authors · 2024-06-05
> > > > > **We are happy to address further questions**
> > > > >
> > > > > Thank you for your detailed and constructive feedback on our manuscript! We deeply value your time and effort in reviewing our work. We are committed to thoroughly addressing your inquiries to enhance the clarity and quality of our submission. We welcome any further suggestions that might contribute to improving our manuscript and increasing its score.

---

### Official Review · Reviewer_pctD · 2024-05-11

**Rating:** 5
**Confidence:** 4
**Ethics Flag:** 1

**Summary:**

This paper proposes a sequence parallal strategy for training long-context LLMs. By carefully optimizing the overall pipeline, the proposed method achieves relatively better performance than previous sequence parallal strategies such as Magatron-LM and Deepspeed Ulysses. Experiments are performed on language modeling tasks with LLAMA 7B.

**Questions To Authors:**

As above.

**Reasons To Accept:**

At first glance, the paper appears to be well-written. The performance is exceptional, and it addresses the limitation of Megatron-LM sequence parallel, where the number of GPUs should be less than the number of attention heads. Moreover, it has outperformed the speed of ring self-attention.

**Reasons To Reject:**

There are several limitations of this paper.

1. The author argues that the proposed method is memory-efficient attention. However, there is no memory comparison in this paper. It shows that it can train 2-8x longer sequences than Ring Self-Attention, which may reflect that it consumes smaller memory. However, when compared with other sequence parallal methods, there is no direct comparison. I would also recommend the author to provide theoritical memory comparison against other methods.

2. It is important to verify the accuracy of distributed algorithm implementations, as there is a risk of bugs causing the optimization to get stuck in local minima. Without providing accuracy verification experiments, it is difficult to justify the correctness of the implementation. Therefore, verifying the accuracy of the implementation is necessary to ensure its reliability.

3. The proposed method only verifies on Llama 7B and its variants. For a general method, it is necessary to test it on other structures or tasks. For example, how will this method perform on bidirectional language modeling?

4. The proposed method only tests on 7B model sizes. It is unclear whether the method will work for smaller or larger model sizes.

5. The conducted experiments were limited to a system with only two nodes. It remains uncertain whether the performance of the system will improve or decline with an increase in the number of nodes, as compared to the existing methods. Further research must be conducted to investigate the impact of scaling on the performance of the system, and to ascertain the optimal configuration for the network.

---

> ### Author Rebuttal · Authors · 2024-05-31
>
> We thank reviewer pctD for the helpful questions.
>
> Q1: Memory comparison with other baseline methods.
>
> A1: We thank the reviewer for raising this question. Our method is memory-efficient mainly because we retained the I/O awareness of FlashAttention when extending it to distributed version. Therefore, we note our method is equivalently memory efficient compared with Megatron-LM(With FlashAttention integration), DeepSpeed-Ulysses(With FlashAttention integration),, and Ring Attention. However, these methods, including ours, are more memory efficient than pipeline parallelism and Ring Self-Attention, because pipeline parallelism introduce imbalance memory consumption leading to a high pressure on the first stage and Ring Self-Attention is not I/O-aware, consuming much more memory due to materialization of the large softmax matrix. The memory efficiency could be empirically reflected by the longest sequence length, as shown in Table 2 and Table 3, that pipeline parallelism and Ring Self-Attention will support a shorter length than our method.
>
> Q2: Accuracy verification of the implementation.
>
> A2: We agree with the reviewer that accuracy verification is necessary. In fact, we have verified the numerical difference between our implementation and the pytorch implementation, which was the same test script used in FlashAttention when they developing the kernel. The test function is included in our submitted codes (`test_op()` function in distFA_async_attn.py). Therefore, we think this proof could provide sufficient guarantee leading to the same accuracy. We note that our method only changes the system implementation for higher efficiency, which does not impact the accuracy and convergence.
>
> Q3: Test with smaller or larger models.
>
> A3: We thank the reviewer for raising this suggestion. We are actively finding more resources to supplement this experiment and we will update the new results once we have additional compute.
>
> Q4: Test with more nodes.
>
> A4: We thank the reviewer for providing the suggestion! Unfortunately, as a research lab, we have very limited access to computational resources, so we cannot find more nodes in a short time to supplement this experiment. However, we will keep finding and update the results once we obtain enough nodes to run.

---

> ### Author Response · Authors · 2024-06-05
> **We are happy to address further questions**
>
> Thank you for your detailed and constructive feedback on our manuscript! We deeply value your time and effort in reviewing our work. We are committed to thoroughly addressing your inquiries to enhance the clarity and quality of our submission. We welcome any further suggestions that might contribute to improving our manuscript and increasing its score.

---

### Official Review · Reviewer_EkNX · 2024-05-14

**Rating:** 6
**Confidence:** 3
**Ethics Flag:** 1

**Summary:**

DISTFLASHATTN tackles the issue of load imbalance encountered during the training of causal language models when the input sequence is distributed across multiple worker GPUs. Typically, this distribution results in later GPUs undertaking a disproportionate amount of computation, as they must attend to a larger segment of the input sequence. DISTFLASHATTN addresses this by redistributing some of the computational tasks from later workers to earlier ones, which are underutilized in the initial phases. This strategy enhances overall computational efficiency as evaluated in various settings. This work also introduces a checkpointing trick to avoid waste of computation.

**Reasons To Accept:**

- Causal training of Transformers holds significant importance. The issue addressed is both practical and valuable. Furthermore, the proposed method demonstrates superior performance in the GQA setting (see Section 4.1, Table 1). Given that GQA is utilized in LLaMa3 training, this methodological improvement is notably beneficial.

- Removing the dependency on the number of attention heads facilitates the utilization of a broader range of devices.

**Reasons To Reject:**

-  The section on related work is inadequately written. It lacks comprehensive discussions on directly relevant works like Ring attention and Ulysses. Essential concepts and several critical references are missing e.g. [1].

- Two primary techniques appear to be straightforward and may lack sufficient novelty.

- I find the experimental setup involving irregular attention heads somewhat untenable for several reasons. First, the use of LLaMa33H appears unrealistic. Although the author cites previous works that employed an irregular number of heads, recent findings by [2] suggest that such configurations can be altered without adversely affecting performance. Consequently, I see no justification for adopting these unconventional head counts. Second, the fourth experimental setting seems far-fetched. According to [3], adjustments in this setting should involve increasing the number of layers rather than merely widening the MLP, as using more layers typically reduces training throughput. However, the author deviates from this approach, opting to increase the MLP width instead. Such comparisons may appear simplistic and are unlikely to be applicable in practical scenarios.

- The evaluation lacks comparisons on moderate sequence lengths against stronger baselines. For instance, it is notable that comparisons involving Ulysses at training lengths of 4K and 8K are absent. Considering the relevance and significance of 8K training in current contexts, its omission is a critical oversight.


[1] Striped Attention: Faster Ring Attention for Causal Transformers: https://arxiv.org/abs/2311.09431

[2] The Case for Co-Designing Model Architectures with Hardware https://arxiv.org/abs/2401.14489

[3] Multi-head or single-head? an empirical comparison for transformer training. arXiv preprint arXiv:2106.09650, 2021.

---

> ### Author Rebuttal · Authors · 2024-05-31
>
> We thank the reviewer EkNX for the questions.
>
> Q1: Lacking discussion of the comparison with RingAttention, DeepSpeed Ulysses, and Striped Attention.
>
> A1:
> - RingAttention: We both distribute FlashAttention along sequence dimension, but our system has two additional innovations: (1) Load-balancing optimization for causal modeling (2x less computation)  (2) Rematerialization-aware gradient checkpointing (1.3x end-to-end speedup).
>
> -  DeepSpeed Ulysses: DeepSpeed Ulysses partitions on the attention heads, and cannot support sequence parallelism degree beyond the head numbers. Meanwhile, we show our method is 1.26-1.88x faster than DeepSpeed Ulysses.
>
> - Striped Attention: Striped Attention is a concurrent work which also found the issue with imbalanced workload. Our method differs from this work by changing the scheduling instead of the order of tokens.
>
> Q2: Two primary techniques appear to be straightforward and may need more novelty.
>
> A2: We agree that the techniques of computation-communication overlapping and load-balancing scheduling are not new. However, we think the application of these ideas in our specific context is novel: (1) identifying the workload imbalance problem in casual language modeling and (2) designing the specific schedule of the asynchronous computation-communication overlap in the distributed attention computation.
>
> In addition, we indeed made more contributions besides these two. We identified the computation overhead when applying gradient checkpointing with FlashAttention. Our rematerialization-aware gradient checkpointing saves 1 forward computation for free, which could lead to speedup in applications of a wide span of applications.
>
> Q3: Justification for adopting unconventional head counts and widening MLP.
>
> A3: We picked the unconventional head counts to highlight the flexibility of our method and can be applied on any model configuration efficiently while other baselines cannot. Regardless of this setting, our method is still faster than other baselines on more popular setup.
>
> Q4: Comparison with DeepSpeed Ulysses on shorter sequence lengths.
>
> A4: On a shorter 8K sequence length, we obtained 1.12x speedup over Megatron-LM while DeepSpeed obtained 1.23x. DistFlashAtten may not be the best choice for shorter context below 8K tokens, where it is hard to overlap the communication overhead inside computation. However, DistFlashAttn performs better than all baselines starting from 16K, which is the main focus of this paper.

---

> ### Author Response · Authors · 2024-06-05
> **We are happy to address further questions**
>
> Thank you for your detailed and constructive feedback on our manuscript! We deeply value your time and effort in reviewing our work. We are committed to thoroughly addressing your inquiries to enhance the clarity and quality of our submission. We welcome any further suggestions that might contribute to improving our manuscript and increasing its score.

---

### Official Review · Reviewer_8Cac · 2024-05-15

**Rating:** 7
**Confidence:** 4
**Ethics Flag:** 1

**Summary:**

The paper titled introduces a distributed variant of FlashAttention, aimed at training large language models (LLMs) with long sequence contexts more efficiently on distributed systems. The paper presents three main innovations: token-level workload balancing, overlapping key-value communication, and rematerialization-aware gradient checkpointing. These techniques collectively enhance the efficiency of memory usage and reduce computational and communication overheads significantly.

The key contributions are:
1. allows for up to 8x longer sequence processing and achieves speedups between 1.24x and 5.64x compared to existing models like Ring Self-Attention and Megatron-LM with FlashAttention.
2. introduces new methods to balance workload across processors, overlap communication with computation to hide latency, and optimize gradient checkpointing to avoid unnecessary recomputation.
3. Extensively evaluated across various setups and model configurations, demonstrating significant improvements in speed and efficiency.

**Ethics Concerns Details:**

no ethics concerns

**Reasons To Accept:**

1. It introduces significant innovations such as load-balanced scheduling, overlapping of communication and computation, and a novel rematerialization aware gradient checkpointing strategy. It critical challenges in training large language models (LLMs) with long sequences, notably improving scalability and efficiency on distributed systems.
2. 8x longer sequences and up to 5.6x speedups are achieved with the proposed techniques
3. Code and implementations are available in the open source

**Reasons To Reject:**

1. Since the proposed techniques are claimed to be more memory efficient. More direct comparisions with Ringattention and other baselines one the memory consumption would be important.
2. The evaluation mainly focus on llama 7 b models, more discussions on how the proposed methods can scale to larger size of model or even MoE models would be interesting.
3. More discussions on how the proposed method can benefit CPU inference.

---

> ### Author Rebuttal · Authors · 2024-05-31
>
> We thank Reviewer 8Cac for highlighting the innovations of our load-balanced scheduling, overlapping of communication and computation, and rematerialization aware gradient checkpointing strategy. We would love address the reviewer’s concerns as below:
>
> Q1: Memory consumption comparison with baselines.
>
> A1: We thank the reviewer for raising this question. Our method is memory-efficient mainly because we retained the I/O awareness of FlashAttention when extending it to distributed version. Therefore, we note our method is equivalently memory efficient compared with Megatron-LM(with flash attention integration), DeepSpeed-Uylesses(with flash attention integration), and Ring Attention. However, these methods, including ours, are more memory efficient than pipeline parallelism and Ring Self-Attention, because pipeline parallelism introduce imbalance memory consumption leading to a high pressure on the first stage and Ring Self-Attention is not I/O-aware, consuming much more memory due to materialization of the large softmax matrix. The memory efficiency could be empirically reflected by the longest sequence length, as shown in Table 2 and Table 3, that pipeline parallelism and Ring Self-Attention will support a shorter length than our method.
>
> Q2: Results on larger models or even MoEs.
>
> A2: We thank the reviewer for raising this suggestion. We would love to have evaluated our method on larger models and other architectures if we had enough resources. We are trying to find more resources to supplement this experiment and we will update the new results once we have additional compute.
>
> Q3: How the proposed method can benefit CPU inference.
>
> A3: We thank the reviewer for asking. Our method is implemented for GPU computation, as we assume that most practice of long-context training and inference will be conducted on GPUs, while our method can also be extended to CPU use case as it is a general tiling optimization. Despite our experiments mainly show the speedup for training, we note that our method can accelerate both the forward and backward of distributed FlashAttention, leading to a speedup for both training and inference.

---

> > ### Comment · Reviewer_8Cac · 2024-06-06
> >
> > Thanks the authors for providing the responses on the more information of memory consumption, MoE and CPU inference. I'd keep the original score.

---

> ### Author Response · Authors · 2024-06-05
> **We are happy to address further questions**
>
> Thank you for your detailed and constructive feedback on our manuscript! We deeply value your time and effort in reviewing our work. We are committed to thoroughly addressing your inquiries to enhance the clarity and quality of our submission. We welcome any further suggestions that might contribute to improving our manuscript and increasing its score.

---

### Decision · Program_Chairs · 2024-07-10

**Decision:**

Accept

**Comment:**

The paper proposes three techniques to do distributed attention on multiple devices, building on FlashAttention and Ring Attention: (1) load balancing in the causal case to avoid devices idling (2) overlapping computation and communication (3) gradient checkpointing that avoids redoing the forward pass of Flash Attention.
These three techniques are evaluated empirically, showing 1.3-2.0x speedup against Megatron-LM with Attention and Ring Attention.
+ The three techniques are conceptually simple and easy to understand
+ The empirical evaluation is comprehensive and convincing
+ Good ablation studies to understand the effects of each of the proposed techniques.
- Techniques (1) and (2) are relatively standard in ML systems. However, demonstrating that these bring speedup to this important problem is still valuable.
Some reviewers have concern over the memory efficiency and scaling of the proposed method. The memory efficiency issue has been addressed in the rebuttal, and scaling to larger model is straightforward.